# A Facile and Controllable Vapor-Phase Hydrothermal Approach to Anionic S^2−^-doped TiO_2_ Nanorod Arrays with Enhanced Photoelectrochemical and Photocatalytic Activity

**DOI:** 10.3390/nano10091776

**Published:** 2020-09-08

**Authors:** Hai Yu, Miao Zhang, Yanfen Wang, Haocheng Yang, Yanmei Liu, Lei Yang, Gang He, Zhaoqi Sun

**Affiliations:** 1School of Physics & Materials Science, Anhui University, Hefei 230601, China; b18101003@stu.ahu.edu.cn (H.Y.); zhmiao@ahu.edu.cn (M.Z.); wangyanfenyu@163.com (Y.W.); y_haocheng@163.com (H.Y.); lym@ahu.edu.cn (Y.L.); hegang@ahu.edu.cn (G.H.); 2Energy Materials and Devices Key Lab of Anhui Province for Photoelectric Conversion, Anhui University, Hefei 230601, China; 3School of Materials Science and Engineering, Anhui University of Science and Technology, Huainan 232001, China; 4Department of Chemistry and Materials Engineering, Hefei University, Hefei 230601, China; ylei531@163.com

**Keywords:** anionic S^2−^-TiO_2_ nanorod arrays, vapor-phase hydrothermal, photocatalytic activity, absorption of visible light

## Abstract

Anionic S^2−^-doped TiO_2_ nanorod arrays (S^2−^-TiO_2_) were synthesized by a facile and controllable vapor-phase hydrothermal (VPH) approach based on the sulfur source of H_2_S gas. After the VPH treatment of TiO_2_ nanorod arrays (TNA), the isolated O^2−^ species replaces the S^2−^ ion in TiO_2_ (TiO_2−x_S_x_). The structural, morphological, optical, compositional, photocatalytic and photoelectrochemical (PEC) properties of the obtained samples were investigated in detail. It was found that S^2−^-TiO_2_ can enhance the separation rate of electron–hole pairs, improve the absorption of visible light, and augment the photocatalytic and photoelectrochemical properties. Anionic S^2−^ doping can significantly adjust the absorption cut-off wavelength (409.5–542.5 nm) and shorten the bandgap (3.05-2.29 eV) of TNA. For the degradation of methylene orange (MO) under mercury lamp light, the 0.24 At%S^2−^-TiO_2_ (0.24S^2−^-TiO_2_) sample exhibited the best photogradation efficiency of 73% in 180 min compared to bare TiO_2_ (46%). The 0.24S^2−^-TiO_2_ showed the highest photocurrent of 10.6 μA/cm^2^, which was 1.73 times higher than that of bare TiO_2_ (6.1μA/cm^2^). The results confirmed that the visible light absorption, photocurrent and photocatalytic activity optimization of TNA are closely related not only to anionic S^2−^-doped but also different ratios of anionic S^2−^-doped. It is noteworthy that the VPH approach is very promising for applications in low cost and highly efficient ion doping into nanomaterials for energy devices.

## 1. Introduction

Since the 1970s, titanium dioxide (TiO_2_) has been recognized as a promising solar-driven photocatalyst due to its high availability, efficiency, excellent functionality, long-term stability, nontoxicity and low cost [1]. However, there are two main problems in the application of TiO_2_: the low separation efficiency of photoinduced electron–hole pairs and the wide bandgap (3.0 eV for rutile and 3.2 eV for anatase). It can only use the ultraviolet region of solar light and photocatalytic activity is limited [1,2,3]. Introducing exterior element doping into material structures has been proven to be an effective approach. Therefore, many reports have indicated that using various ions (C, N, F, S, Fe, Co, Ag, Ni) doped into TiO_2_ narrows the bandgap of TiO_2_ and improves the photocatalytic activity of TiO_2_ under visible light [4,5,6,7,8]. Seed growth, chemical vapor deposition (CVD), hydrothermal and sol-gel methods synthesize various element-doped TiO_2_ common routes. However, transition (Fe, Co, Ag, Ni) metal-doped TiO_2_ has a high electron–hole recombination rate [7,8]. To overcome that disadvantage, non-metal-doped TiO_2_ has been synthesized by intensive efforts [9]. Sulfur is one of the most common non-metal elements, and is easy to synthesize, has a low manufacture cost and is easily industrialized. By theoretical calculation, anionic S has a larger ionic radius compared to O, N and F, which visibly modifies the electronic structures of TiO_2_ and narrows the TiO_2_ bandgap. Umebayashi et al. synthesized S-doped TiO_2_ for the photocatalytic degradation of methylene blue in visible light [10,11,12]. Interestingly, unlike other elements, S tends to have two species (cations S^4+^/S^6+^ and anionic S^2−^). The occurrence of an anionic or cationic state is considered to be dependent on the location in TiO_2_: in substitution for either O or Ti atoms [2,11]. Yamamoto et al. reported their theoretical results on S-doped rutile TiO_2_, showing the necessity of experiments to investigate different chemical species of S-doped TiO_2_ [13,14].

In previous experimental works on S-doped TiO_2_, cations S^4+^ (Ti_1−__2__x_O_2_S_2__x_) and S^6+^ (Ti_1−__3__x_O_2_S_2x_) were used instead of Ti^4+^, which shows a slight optical absorption shift from the ultraviolet region to the visible light region (near 40 nm) and the optical effect was not obvious [13,14,15,16,17]. The first principle calculations revealed that anionic sulfur (S^2−^)-doped TiO_2_ shows completely different spectral behavior compared to that of bare TiO_2_ and of cationic S^4+^/S^6+^-doped TiO_2_. Anionic S^2−^-doped TiO_2_ has a broad absorption spectrum (400–700 nm), which greatly reduces the bandgap and improves the photocatalytic activity under visible light [2,12]. This is due to new occupied and strongly localized electronic states which are formed in the gap, the anionic S3p electron orbitals form shallow impurity levels in the bandgap, located about 0.9–1.6 eV above the TiO_2_ valence band [13].

While experiments have confirmed that both (anionic or cationic) states of S ions are simultaneously doped into TiO_2_ nanotubes [2], only one chemical species of anionic S^2−^-doped TiO_2_ has rarely been observed in the experimental stage. To this extent, the main reason is that a single anionic S^2−^ source is not easy to synthesize in the conventional experiment. H_2_S is a good source of S^2−^. In this work, we speculate that the vapor-phase hydrothermal (VPH) process could be superior to that of conventional liquid-phase hydrothermal (LPH) processes for collecting H_2_S gas and providing a single anionic S^2−^ source for the experiment [17,18,19,20]. Compared with the traditional LPH method, all the reactions are triggered by volatile reactants and reflect on the substrate surface [18]. The VPH process differs dramatically from LPH processes in which the chemical forms and available concentrations of substance can be affected by many factors (solution composition, pH and the concentration), yet these factors hardly affect the VPH process [17]. Under VPH conditions, to achieve high concentrations of anions, S^2−^-doped TiO_2_ is easier to implement and more operationally controllable compared to the LPH process [19,20]. Until now, it has rarely been reported that only one chemical species of anionic S^2−^-doped TiO_2_ can shorten the optical bandgap and promote the utilization of the solar energy spectrum. In this paper, only one species of anionic S^2−^ (not cations S^4+^ and S^6+^) -doped TNA is observed, which is different from the previous works about the introduction of induced S (S^2−^, S^4+^, S^6+^)-doped TiO_2_ nanotubes. In order to realize only one species of anionic S^2−^-doped TiO_2_, TNA with tiny size is selected by as the precursor, whose special structure made it possess a high reaction activity and facilitate the diffusion of S^2−^ into TiO_2_ by the VPH method.

## 2. Experimental

### 2.1. Preparation of Anionic S^2−^-TiO_2_ Nanorod Arrays

MaterialsTetrabutyl titanate C_16_H_36_O_4_Ti, 37% hydrochloric acid (HCl), Ferrous sulfide (FeS), deionized water and absolute ethanol were used for the sample preparation. All the chemicals and reagents were of analytical grade and used without further purification.

Using C_16_H_36_O_4_Ti (TBT) as the Ti source, H_2_S, prepared from FeS and hydrochloric acid (HCl), was used as an anionic S^2−^ source, S^2−^-TiO_2_ was prepared by VPH. The S^2−^-doped TNA was prepared using a two-step method. First, the TNA was fabricated using a simple hydrothermal method. In detail, 15 mL HCl (10 mol per liter) and 15 mL deionized water were mixed in a 50 mL beaker, followed by the addition of 0.5 mL of TBT. The mixture is stirred for 20 min, then transferred to a teflon-lined autoclave (50 mL). Pre-cleaning fluorine-doped tin oxide film (FTO) was used as a substrate (1.5 cm × 3 cm) and the mixture was injected into a polytetrafluoroethylene-lined cylindrical autoclave. The TNA was formed by heating for 12 h at 150 °C [3]. Secondly, the TNA was in situ doped in the VPH reactor (about 150 mL teflon-lined autoclave), where H_2_S gas was produced slowly by FeS and HCl reaction [21,22]. The chemical processes of FeS and HCl are as follows:(1)$$\mathrm{Fes}+2\mathrm{HCl}={\mathrm{FeCl}}_{2}+H_{2}S↑$$

This formula is used because it can yield single H_2_S gas from a cheap source of FeS; although it can be performed at room temperature, in order to improve the efficiency of H_2_S production, FeS and HCl were placed in the VPH reactor where the TNA, on FTO substrate, was held above the reactants. The overall formation process of S^2−^-TNA was realized by VPH as illustrated in Scheme 1 and Figure 1 (Supporting Information) [17].

As the reaction proceeded, H_2_S was used as anionic S^2−^ precursor, and a part of the O^2−^ ions in the rutile TiO_2_ crystal are replaced by S^2−^ ions and ultimately S^2−^-TiO_2_ formation, as shown in Figure 1.

In the VPH processes, TNA used as a substrate, as it is well known that the gas concentration provided for VPH treatment essentially drives the interaction between the original TiO_2_ crystal and H_2_S gas [17]. Therefore, the concentration of H_2_S was studied in the second aspect. The VPH treatments were carried out at different masses of FeS (2.2, 4.4, 6.6 g) and volumes of HCl (5, 10, 15 mL) in a duration (12 h for 260 °C). Unfortunately, when there is a mass of 8.8 g FeS and 20 mL of HCl, a large amount of HCl liquid boils at 260 °C, causing the TNA to be destroyed and the anion doping cannot be achieved. By theoretical calculations, (2.2, 4.4, 6.6, 8.8 g) the FeS complete reaction under standard conditions (0 °C, 100 kPa) produced 560 mL, 1120 mL, 1680 mL, and 2240 mL of H_2_S gas, respectively. The H_2_S gas produced is much larger than the volume of the VPH reactor (150 mL), resulting in an increase in the internal pressure, which is beneficial to replace O^2−^ with S^2−^ ions. The duration of the VPH was set to 12 h and the aim is to make VPH react to completely dope at 260 °C. The VPH reaction was carried out at 260 °C for 12 h, then the autoclave was removed from the oven and allowed to cool to room temperature. Before optical testing or structural characterization, the S^2−^-TiO_2_ was washed with deionized water to remove the sulfur produced during the VPH treatment.

### 2.2. Characterizations

The microstructures of samples were studied by an X-ray diffractometer (XRD, M18XHF, Tokyo, Japan) with Cu_k__α_ radiation (λ = 0.154 nm) operated at 40 kV and 30 mA. The morphological and structural information of samples was characterized using field emission scanning electron microscopy (SEM, Hitachi-S4800, Dallas, TX, USA), a transmission electron microscope (TEM, HITACHI, Tokyo Japan) and the attached energy dispersive X-ray (EDX) spectroscopy. The chemical composition and quantitative investigation of chemical states of various elements in films is performed by X-ray photoelectron spectroscopy (XPS, Ulvac-Phi 5000 Versaprobe, hermo-VG Scientific, West Sussex, UK). The measurement absorption intensity (by direct transmittance measurements) and calculation of the bandgap of the samples were characterized by an ultraviolet-visible spectrophotometer (UV-vis, Shimadzu, Shanghai, China).

### 2.3. Photoelectrochemical and Photocatalytic Measurements

Photoelectrochemical (PEC) properties of the samples were tested on an electrochemical workstation (CHI 660D, Chenhua, Shanghai, China). Na_2_SO_4_ aqueous solution (0.1 M) was used as the electrolyte, and a 150 W Xe lamp (150 A, Zolix, Shanghai, China) was used as the working electrode of light sourcing. The optical responses of different samples were measured by a 20s optical switch period with a bias voltage of 0 V.

A diagram of the photocatalytic experimental setup is illustrated in Scheme 2. The photocatalytic properties of the samples were evaluated by monitoring the photodegradation of methylene orange (MO) under simulated solar light at ambient atmospheric pressure and temperature. The photodegradation efficiency of the samples was tested by simulating sunlight exposure (full spectrum). A 25 W mercury lamp was used as the full spectrum source. In the test, the sample (1 cm × 1.5 cm) was immersed in an oval beaker of 10 mL MO solution. Before the test, the dye solution was kept in the dark for 30 min to achieve an equilibrium of adsorption and desorption between the sample and the dye. The distance between the lamp and the sample is maintained at approximately 10 cm. At the same intervals, 5 mL of the MO solution was collected and the concentration of the MO solution was analyzed using a UV-vis spectrophotometer (UV-vis, Shimadzu, Shanghai, China). At the end of each test, the solution was carefully returned to the reaction beaker and the reaction was continued until the degradation of a sample was almost complete.

## 3. Results and Discussion

### 3.1. Microstructure Analysis

Figure 2 shows the XRD patterns of TiO_2_, 0.07 At%S^2−^-TiO_2_ (0.07S^2−^-TiO_2_), 0.11 At%S^2−^-TiO_2_ (0.11S^2−^-TiO_2_) and 0.24 At%S^2−^-TiO_2_ (0.24S^2−^-TiO_2_) samples. XRD patterns were recorded for the TiO_2_ with different S^2−^-doped percentages. Figure 2a shows XRD data of the samples before and after VPH doping, showing similar diffraction patterns, which may be attributed to the TiO_2_ crystal structure of rutile phase. Diffraction dominating peaks centered at 2θ = 36.0°, 41.2°, 54.3°, and 62.7° are ascribed to (101), (111), (211) and (002) planes of rutile TiO_2_ (JCPDS 21-1276), respectively. However, the sulfur diffraction peak cannot be detected in the XRD pattern, which is mainly due to the strong diffraction intensity of rutile TiO_2_ and the low content of sulfur ion. These favorable results indicate that VHP can hardly change the crystal structure. In addition, the slight displacement of the diffraction peak after VPH doping indicates an increase in the lattice strain due to the addition of sulfur to the lattice. The calculation method of crystal surface spacing is based on the Bragg equation;
(2)2dsinθ=nλ
where d is the spacing between crystal planes, 2θ is the diffraction angle, λ is the wavelength fixed by the instrument (λ = 0.154 nm), this formula asserting that the smaller the angle, the bigger the d. In geometry, the direct effect of replacing O with a larger radius of S ion should be the expansion of TiO_2_ crystal planes. By doping anionic S^2−^ ions with different ratios into TNA, the diffraction peak is slightly shifted to a low angle compared with the original TNA, due to the radius of the sulfur ion being larger than the radius of the oxygen ion (S^2−^ = 2.05 Å and O^2−^ = 1.44 Å) [23,24]. Figure 2b shows the S^2−^-doped TNA diffraction peaks being slightly shifted, which is mainly realized in (101) peaks. The calculation results show that the expansion of unit plane spacing caused by S-doping exists regardless of the doping level and the XRD spectral line shifts to a smaller angle more obvious with the increase of doping ratio. Interestingly, noteworthy is that the half-width peak was getting bigger, especially in (101), as shown in Figure 2b. In fact, there may be two reasons for the widening of the diffraction peak: the grain size is small for a moment, and there is a “microscopic strain” inside the grain. Here, it may be the latter, which indicates that perhaps the anionic S^2−^-doped TiO_2_ have “microscopic strain” of the lattice. XRD results confirm that the structure of TNA, in the rutile phase of these samples, did not change significantly, but there is an obvious doping phenomenon.

To gain insight into the morphological and microstructures of the as-synthesized TNA and S^2−^-TiO_2_, samples were performed using field emission scanning electron microscopy (SEM) and transmission electron microscope (TEM), as shown in Figure 3 and Figure 4.

Figure 3a,b show a top view of the 0.24S^2−^-TiO_2_ and bare TNA. It is evident that TiO_2_ has a well-aligned nanorod array structure in the FTO, with an average diameter of 300 nm. Figure 3c shows a cross-section view of the 0.24S^2−^-TiO_2_ with a height of about 2.6 μm. Regarding the SEM images of samples before and after VPH treatment, the results indicated that the TNA treated with VPH has hardly significant structural or morphological changes, which is consistent with the XRD results. In addition, according to the EDX analysis in Figure 3d, the weight ratio of C, O, S, and Ti was determined to be 9.05%, 59.28%, 0.31%, and 29.89%, respectively. TEM and High-Resolution TEM (HRTEM) images in Figure 4 further indicate the morphology and structure of S^2−^-doped into TNA. Figure 4a shows the side view of a TiO_2_ nanorod, approximately 100 nm in diameter. The HRTEM images in Figure 4b,c clearly show two kinds of lattice fringes of bare TNA and 0.24 At%S^2−^-TiO_2_. It is estimated that the stripe spacing on the same side of TiO_2_ is different, and the crystal plane spacing of these two structures could be 0.28 nm and 0.31 nm, respectively, as shown in Figure 4e. The HRTEM images of samples before and after VPH treatment reveal that some large radius ions may dope into the TiO_2_ nanorod, resulting in weak lattice distortion. This phenomenon shows that a small part of ion substitution may effectively change the electron orbit and ensure the improvement of optical properties. Figure 4d shows the electron diffraction diagram (SAED) in the selected region, further verifying the TiO_2_ nanorod array by VPH can not be changed by the single crystal properties of a nanorod [3]. Similarly, Figure 4f analyzed EDX of the samples after VPH under TEM image, and the weight ratio of O, S, and Ti was determined to be 36.10%, 0.35%, and 55.26%, respectively. The EDX data of TEM and SEM image analysis are essentially the same.

### 3.2. XPS Analysis

XPS was used to further investigate the electronic state of the present elements in the near-surface region (about 10 nm). The chemical states and content of elements in TNA were analyzed by XPS, as shown in Figure 5.

For comparison, the XPS spectra of the TiO_2_ and the 0.24S^2−^-TiO_2_ samples were analyzed. Figure 5a shows a wide-scan survey spectrum, S, O, Ti and C in the 0.24S^2−^-TiO_2_ sample and O, Ti, and C in the TiO_2_ sample can be clearly observed. It is proved that sulfur is doped into TNA by VPH.

The application of XPS in the quantitative analysis of elements is that the intensity signal of the measured spectral line is transformed into the content of elements, converting the area of the peak to the concentration of the corresponding elements. At present, the most widely used method is the element sensitivity factor method for quantitative analysis, using the formula:(3)ninj=AiSiAjSj
where *A* and *S* represent the integral area of curve and sensitivity factors (C1s:1; O1s:2.88; S2p:1.88; Ti2p:6.47). According to the integral area of Figure 5b curve, the *A* (I × eV) of each element is calculated, as shown in Table 1.

After calculation, the results are shown in Table 2.

This shows that S was doped, and the TNA and S ion doping rate increases with the increase of FeS by VPH.

The calculations indicate that the ratio of O atom decreases and S gradually increases with the increase of H_2_S concentration. This conclusion implies that the S substitutes for a small number of O atoms. The main reason is that the thermal energy provided in the VPH treatment essentially drives the interaction between the bare TNA crystals and the H_2_S gas. However, the low ratio of S is due to the radius of the S^2−^ ions, which is relatively large, and only a small amount of O ions can be replaced by S ions in the TiO_2_ crystal (O^2−^ = 1.44 Å and S^2−^ = 2.05 Å) [25,26]. The weak change of the lattice size of TiO_2_ is caused by the substitution of O by S, which is a good explanation for the small distortion spectrum of XRD. Figure 5d shows the oxidation state of the S atoms incorporated into the TiO_2_ particles, which is determined to be mainly S^2−^ from XPS spectra. The XPS peak appeared in the region with the binding energy of 160–163 eV at the S2p nuclear level, which was believed to be due to the Ti-S bond formed when O^2−^ was replaced by S^2−^, and the presence of S^6+^ and S^4+^ made binding energy peak in the range of 167–170 eV [27,28,29]. This result confirms that only one chemical state anionic S^2−^ was successfully doped into TNA, which is different from the previous works about S-doped TiO_2_ that contained cationic S^4+^ and S^6+^ [30].

### 3.3. Optical Properties of S^2−^-TiO_2_ Nanorod Arrays

The absorption intensity and calculation of the bandgap of the samples were characterized by a UV-vis spectrophotometer, as shown in Figure 6. The absorption intensity was obtained from direct transmittance measurements.

Compared to the bare TiO_2_, it was observed that S^2−^-TiO_2_ had obvious red shift and enhancement of visible light absorption, as shown in Figure 6a,b. In this paper, the intrinsic absorption intensity (IA) is defined as the integral area under the sample absorption curve from 400 nm to the cut-off wavelength range. It is estimated that the IA (I × nm) of 1.5, 37.5, 52.8, 77.4 corresponds to 0.00, 0.07, 0.11 and 0.24 doping anionic S^2−^ levels, respectively, as shown in Figure 6b. The visible light absorption intensity (VA) is defined as the integral area under the sample absorption curve in the visible light region (400–700 nm). It is estimated that the VA (I × nm) of 7, 51, 75, 142 corresponds to 0.00, 0.07, 0.11 and 0.24 doping anionic S^2−^ levels, respectively, as shown in Figure 6b. Figure 6a,c show S^2−^ doping can significantly adjust the absorption cut-off wavelength from 409.5 nm to 542.5 nm and the bandgaps also decreased from 3.05 to 2.29 eV. Figure 6b also shows that intrinsic absorption values are different at different doping ratios. Interestingly, when the doping ratio is 0.11% and 0.24%, the absorption curve has a relatively weak absorption in the absorption spectrum besides S^2−^-TiO_2_ intrinsic absorption. It can be found that this absorption may be a doping energy level caused by S^2−^ doping. However, many studies have shown the UV-vis absorption optical spectra and bandgap of S^4+^ or S^6+^-doped TiO_2_ (Table 3) [31,32], indicating that S^2−^ doping is more pronounced. The bandgap (E_g_) of the sample can be calculated by the cut-off absorb wavelength (λ) according to the formula: E_g_ = 1240/λ [33]. The result of the cut-off wavelength, absorption intensity of visible light, bandgap theoretical value and the bandgap of each sample are shown in Table 4.

The narrow bandgap and the high visible light absorption intensity suggest that S^2−^-TiO_2_ improves the absorption of visible light, so that the photoelectrochemical and photocatalytic properties can be improved.

Many experimental results and literature show that the cut-off wavelength of cationic S^4+^ or S^6+^-doped TiO_2_ did not change significantly (about 40 nm) [35,36]. It is only theoretically calculated that there is only one chemical species of anionic S^2−^-doped TiO_2_ UV-vis absorption, as few experiments have been verified [13]. Fortunately, in this paper, only one chemical state anionic S^2−^ was successfully doped into TNA by VPH. Favorable experimental values are due to some S^2−^ occupying the replacement of some O atoms on the TNA. The bare TiO_2_ band structure: the valence band maximum (VBM) is mainly composed of the O2p orbit, and the conduction band minimum (CBM) is mainly occupied by the Ti3d state. The S3p state is slightly higher than the upper edge of the bare TiO_2_ valence band (O2p) [36]. Harb et al. calculates the optical absorption of anionic S^2−^ doping in bulk anatase TiO_2_ by first-principle calculations. It has been confirmed that anionic S^2−^-TiO_2_ provides new occupied and strongly localized electronic states in the gap, situated at about 0.9–1.6 eV above VBM of anatase TiO_2_, which causes the absorption red shift [13]. Some of the new electron transitions are from S3p orbit to Ti3d orbit, which results in the broad visible-light absorption and narrow bandgap, as shown in Figure 6d. It is noteworthy that the 0.24S^2−^-TiO_2_ bandgap shortens 0.76 eV, as it is good to verify the theoretical results of the anionic S^2−^-doped TiO_2_.

### 3.4. Photoresponse

The generation and transport efficiency of the excited electron for S^2−^-doped and bare TNA were studied by transient photocurrent response experiments. The optical response performance of the electrode was carried out under the irradiation of simulated sunlight. Figure 7 shows the photocurrent–time characteristics of the S^2−^-doped and bare TNA. It was clear that all photoanodes had a fast and good optical response. Under dark conditions, the photocurrent density is almost zero, while the current density increases immediately after intermittent light is turned on. In particular, the photocurrent of the samples gradually increases with increasing S^2−^. Figure 7 shows the current density of different ratios; S^2−^-doped is about 1.16 times, 1.35 times and 1.74 times higher than that of bare TNA, respectively. This shows more efficient transfer performance of interfacial electrons with an increase in the S^2−^ ratio. The result shows low concentration single-ion doping TNA has an ideal effect on PEC properties.

### 3.5. Photocatalytic Activity Measurements

MO is a common refractory organic pollutant that is not volatile. Its chemical structure belongs to most kinds of azo dyes. Therefore, taking MO as the research object has certain representativeness. With 25 W mercury lamp as the light source, TiO_2_ and different ratios of S^2−^-TiO_2_ samples were a photocatalyst, and MO was the target degradant. The decolorization rate was measured by UV-vis spectrophotometry, which was attributed to 462.2 nm, corresponding to the absorption peak of the azo structure in MO. In each experiment, the samples were immersed in a rectangular quartz cell containing a 12 mL of MO solution (25 mg/L). Before the experiment, different samples were put into MO for dark adsorption for 2 h, so that MO could reach the maximum adsorption equilibrium on the photocatalyst surface. The experimental environment is identical, except for the samples. Then, the photocatalytic reaction was carried out and the degradation efficiency was measured by UV-vis regular sampling. The degradation efficiency of pollution after UV irradiation of OM was calculated based on the relationship between absorbance and concentration,
(4)$$\eta (\%)=\left(C_{0}-C_{t}\right)/C_{0}=\left(A_{0}-A_{t}\right)/A_{0}\times 100\%$$
where C_0_ and A_0_ represent the initial concentration and absorbance, C_t_ and A_t_ represent the concentration and absorbance after t min reaction of the MO at the characteristic absorption wavelength of 464.2 nm.

Figure 8 shows that the photocatalytic activity of samples is evaluated by the degradation of MO under a 25 W mercury lamp. With the increasing exposure time, the characteristic absorption peak of MO at 462.2 nm drastically decreases. Figure 8a–d show photodegradation curves of MO using samples under the 25 W mercury lamp for various durations, respectively. Figure 8e,f show the MO degradation rate of bare TiO_2_, 0.07S^2−^-TiO_2_, 0.11S^2−^-TiO_2_ and 0.24S^2−^-TiO_2_ is about 46%, 54%, 60% and 73% under 25 W mercury lamp irradiation for 180 min, respectively. Compared with bare TiO_2_, the TNA treated by VPH shows better photocatalytic activity. According to the above photocatalytic mechanisms, it can be concluded that the S^2−^-doped TNA resulted in an excellent photocatalytic performance.

To further demonstrate the remarkable effect of S^2−^ doping, we added a filter to a 25 W mercury lamp so that the light source illuminates the sample at a wavelength between 380–800 nm and then compared the effect of sample degradation as shown in Figure 9. Figure 9 shows the degradation curves of MO in TiO_2_ and 0.24S^2−^-TiO_2_ photocatalysts. Due to the wider bandgap, TiO_2_ is stimulated only by UV light (380–400 nm). The MO decomposition rate of bare TiO_2_ was 22.3%, indicating its lower photocatalytic activity. The photocatalytic activity of 0.24S^2−^-TiO_2_ to MO solution (51.8%) was enhanced by doping with S^2−^, indicating that S^2−^ plays a key role in enhancing the photocatalytic activity.

## 4. Conclusions

In conclusion, in this paper, a facile VPH approach was demonstrated in anionic S^2−^ to replace a part of O^2−^ in TNA so that it enhances the photoelectrochemical performance and improves the photocurrent. It was found that the S3p orbit of anionic S^2−^ played a key effect on the properties of TNA. Anionic S^2−^ doping TNA can significantly adjust the absorption cut-off wavelength, expanding to the visible light range from 409.5 nm to 542.5 nm of TNA and reducing the bandgap of TNA to 2.29 eV. Compared with the bare TNA, the VA (7–42) and IA (1.5–77.4) of S^2−^-TiO_2_ samples are greatly enhanced, which has a potential application prospect. In the photodegradation of MO, the 0.24S^2−^-TiO_2_ sample shows the strongest degradation efficiency of 73%, while the degradation efficiency of bare TiO_2_ is 46%. The 0.24S^2−^-TiO_2_ sample also shows the highest photocurrent of 10.6μA/cm^2^, which was 1.73 times higher than that of bare TiO_2_ (6.1 μA/cm^2^). Sample characterization and optical absorption intensity evaluation results demonstrated a close relationship between the S^2−^ dopant level of TIO and their photocatalysis activity and PEC. Interestingly, the 0.24S^2−^-TiO_2_ bandgap shortens to 0.76 eV, well verifying the theoretical results of the anionic S^2−^ doping TiO_2_. In addition, it is few reported that H_2_S, as an anionic S^2−^ source, dopes into TNA by VPH, which has a new broad application prospect in effective and low-cost doping.

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
