# Peer review of "A Facile and Controllable Vapor-Phase Hydrothermal Approach to Anionic S2−-doped TiO2 Nanorod Arrays with Enhanced Photoelectrochemical and Photocatalytic Activity"

_nanomaterials, 2020, doi:10.3390/nano10091776_

Round 1
Reviewer 1 Report
Dear Authors,
I consider the article relatively valuable and well written. However, I have some comments related primarily to the insufficient description of the experimental part regarding the determination of the activity of catalysts used in the MO degradation process. I present them below.
A.
In my opinion, the description and characterization of the method of performing experiments to determine MO degradation efficiency is insufficient. The presented (in paragraphs 2.3 and 3.5) text is not understandable, at least for me. In my opinion, the authors should describe these runs in detail (basic information was provided, but in my opinion, they are insufficient); It means, the authors ought to:
- describe the experimental set-up and the mode of performing them - in which way the experiments were carried out, (batch, semi-batch, continuous, reaction mixture with stirring of it or without stirring etc.),
- provide a diagram/drawing of the experimental set-up,
- specify the type of the reactor - a batch reactor, semi-batch, continuous,?
- determine the atmosphere above the reaction mixture, (inert gas, air, oxygen, gas flow during the run (yes / no)),
- give the reactor capacity, the samples volume,
- give the number of experiments, the number of repetitions of the runs,
- give the results of blank experiments (TIO2 without S2-)
- determine the error (or accuracy) of the analytical determination of MO concentration and the systematic error of the method.
- explain whether decomposition products were analyzed,
- explain whether TOC was analyzed and determined in liquid phase during the experiments,
- explain whether the gas phase composition was analyzed (to determine the concentration of CO2),
- explain whether an elemental carbon balance was made - which allows the assessment and validation of correctness of performing of the MO degradation experiments.
etc.
If the way of conducting experiments and the description of the experimental set-up is already described in detail in another publication of the authors, please indicate this article if it is available.
B.
It would be advantageous to compare your own research results with similar research presented in the literature (for example, graphical comparison or presented in Table).
C.
In my opinion, paragraphs 2.3 and 3.5 are the most important parts of the article, because all TiO2 modifications and the preparation of new types of catalysts make sense only if their catalytic activity is higher than the activity of other known catalysts so far as well as theirs utility properties and environmental impact are better and more advantageous. For this reason, the proof of the catalysts activity - that was conducted in the proper way, consistent with the art and scientific methodology is one of the most important facts. In my opinion, in the presented article, this proof is insufficient.
Reviewer 2 Report
The authors of this work prepared solphur-doped (S-doped) titania (S-TiO2) using a vapor phase hydrothermal (VPH) approach that is supposed to dope the titania only via substitutional doping of oxygen anions by S2- species (anionic doping). The authors highlighted the importance of this point, as the S atoms can assume different oxidation states and hence, S doping can occur via introduction of either S(2-)anions or of S(+4) and S(+6) cations. The VPH procedure allows to preclude the cationic doping and to synthesize doped titania in which the S(2-) species is the only one present at a relevant amount.
Despite the fact that the visible light induced photocatalytic activity of S-doped TiO2 is well known since many years (early reports can be found by Umebayashi, Ohno and other authors since 2003), I think that the fact that
authors were able to study TiO2 doped by a selected dopant species of sulphur (without S cations) makes the work in principle interesting and suitable for publication on Materials.
However, there are several major problems and flaws in the original manuscript, which I point out next. Therefore, I recommend a major revision of the work that shall be reviewed again in its revised version.
Revisions should regard the following points:
1) The photocatalytic experiments and data shall be revised. In fact, the light emission of a mercury lamp typically includes both a continous spectrum of visible light and UV (and visible) emission lines. As we are of course more interested in the photocatalytic activity activated by visible light, it should be clarified whether the photodegradation of the dye has been activated by UV light or by visible light. This is a very important point that is unfortunately not clarified in the manuscript. Therefore, the authors should provide an emission spectrum of their lamp and filter out eventual UV components (for example, filter out all the light at wavelength < 440 nm) when repeating additional photodecoloration experiments. Otherwise, the result would not be much less interesting for applications.
2) The importance of isolating an anionic doping (i.e. S(2-) ions) from a possible cationic doping (S4+ or S6+) shall be highlighted in a persuasive and tangible way. For example: would a cationig doping lead to worst photocatalytic performances? Or to different optical absorption spectra? Is there any advantage in achieve a purely S2- doping? The authors might report experimental investigation on a cationic doped titania or at least spend spart of the discussion comparing their results with some of the literature results reporting about the photocatalytic activity of S(4+) or S(6+) doped titania (such as, for example, the references [13-17] they cite).
Preferably, this shall be done by also mentioning experimental results and not only first-principle works (as the ref 13 they mention). First principles works are ok but they usually do not allow to take into account the whole real phenomenology.
3) The introduction is not very clear. The authors should revise it by separating more clearly two different parts: 1) what is the goal of using the VPH approach, why is important to obtain anionic doping and why VPH should provide it, and next: 2) what exactly has been done in the work. For example: "In this work we managed to obtain S-doped TiO2 doped by only the anioinic sulphur species S2− ." and then state the main results found in terms of
photocurrent and photocatalytic degradation of methyl orange.
4) The photoluminescence results are clearly unreliable. The PL emission of titania does not look at all as the ones reported in the work. First of all, it can be seen that the excitation light in not compltely filtered out. Also, the spectra shall be reported up to 900 nm, as the most interesting characteristics of PL spectra of TiO2 occur from 500 to about 900 nm, considering that the authors obtained TiO2 in rutile phase which has almost no emission in the 500-800 nm range and shall exhibit a large PL peak at about 820-840 nm. Please report new spectral data and extend the measurement in the entire visible andf near-IR range (from 350 to 900 nm).
5) Figure 6 should be improved by using larger axis titles and uniform axis thickness (a difference in axis thichness of Figs 6a and 6b is noticeable).
6) The experimental methods section (2.2) is not very clear when describing the optical absorption measurements. Were the absorption spectra obtained by direct transmittance measurements, or by measuring the diffused reflectivity spectra R and then applying the Kubelka Munk formula F=(1-R)^2/2R ?
7) The equation on page 7 is not readable. Please adjust it.
8) There are several serious grammar mistakes. I report those that I noticed.
Abstract:
Line 16: "the isolated O2-" (not S2-) "species replaces the S2− ion"
Line 19: "it was found THAT S2- - TiO2.."
Line 22 "...THE bandgap..."
Manuscript:
Line 46: "Sulfur is one of the most common non-metal elements, EASY TO SYNTHESIZE..." (not "simple synthesis of ..")
Line 47: "anionic S HAS (not "is") a larger ionic..."
Line 48: Eliminate "since"
Line 51: "The occurrence of an anionic or cationic state is considered to be dependent on..."
Line 55: Eliminate "the" ("In previous experimental works...". Use "on" instead of "about".
Line 58: Eliminate "the", eliminate "only"
Line 59: "compared TO that OF bare TiO2 and OF cationic..."
Line 60: "HAS a broad spectrum" (not "is")
Line 61: Missing space
Line 64: Missing space
Line 65: Unclear. Maybe author meant that both species (anion and cation) are usually present? Please rewrite in a clearer manner.
Line 67: The fact that the H2S is a good source of S2- is not "inspired" by anything. Please revrite the sentence.
Line 68: "WE speculate" (not "it")
Line 72: Missing space
Line 123: "Absorption spectra were determined by..." (explain how they were determined, see point 6 above)
Line 126: "FLUORESCENCE" (not "rescence")
Line 151: "Bragg" (not "bragg", use the capital B).
Line 219: Space missing ("Table 1")
Table 2: Use the capital o (O) for oxigen, i.e. n(O) and n(S) as title of the third and last columns
Line 230: "The calculations indicate that the ratio of O atoms..."
Line 232: "The main reason is THAT THE thermal energy.."
Line 245: "Optical absorption spectra were obtained by ..." (please explain how exactly the spectra were obtained).
Line 226: The expression Eg=1240/l is NOT the Kubelka Munk formula! It is just the standard Planck expression E=hc/l (where h and c are the Planck constant and the speed of light). Please correct.
Round 2
Reviewer 2 Report
The Authors revised several parts of their manuscript. However, there are still some points to be revised:
As I said previously, the photoluminescence data are unreliable. Based on the screenshot the authors sent in the response letter, I can easely see that there are some experimental problems, as it happens frequently when benchtop fluorescence instruments are used to detect very small signals as the photoluminescence of TiO2 (or of other metal oxides). For example, the excitation light scattered by the sample is not completely filtered out, as proved by the presence of the second order diffraction peak at 700 nm in the detected spectrum. Its presence prevents measuring the photoluminescence in the visible range. Moreover, the rutile sample is expected to emit light in the near-infrared (with a peak at about 840 nm) but the spectra reported in the response letter appear to have a peak at 900 nm which is also in conflict with the literature data.
Considering that the photoluminescence data would be in any case not important for the rest of the work, authors should simply remove Fig 7(a) and remove the comments relative to it.
Moreover,
- Remove lines 376-381 (the grammar is wrong and the equations also are)
- Check the affiliations of the authors. The affiliations in the title are three and are indicated by letters ("a","b", "c") but immediately after the affiliations become four affiliations and are indicated by numbers indicated by 1,2,3 and 4.
Round 3
Reviewer 2 Report
Based on the new revisions, it is my opinion that the manuscript is publishable in the present form.